



# Early Morning Peaks in the Diurnal Cycle of Precipitation over the Northern Coast of West Java and Possible Influencing Factors

Erma Yulihastin [1,2], Tri Wahyu Hadi[1], Nining Sari Ningsih[3], Muhammad Ridho Syahputra[1]

[1]Atmospheric Sciences Research Group, Faculty of Earth Sciences and Technology, Institut Teknologi Bandung, Bandung, 40132, Indonesia
[2]Center of Atmospheric Sciences and Technology, National Institute of Aeronautics and Space, Bandung, 40173, Indonesia
[3]Oceanography Research Group, Faculty of Earth Sciences and Technology, Institut Teknologi Bandung, Bandung, 40132, Indonesia

*Correspondence to*: Tri Wahyu Hadi (tri@meteo.itb.ac.id)

**Abstract.** The diurnal cycles of precipitation over the northern coast of West Java have been studied using the Tropical Rainfall Measuring Mission (TRMM) Real Time Multi-satellite Precipitation Analyses (MPA-RT) products with records spanning from 2000 to 2016, emphasis on the occurrences of early morning precipitation peaks. Diurnal precipitation over the study area during November to March is basically characterised by precipitation peaks that occur in the afternoon to evening time (1500–2100 LT) but secondary peaks in night to morning time (0100–0700 LT) are also pronounced in January and February. Harmonic analysis method was then applied on data of January and February to objectively determine the diurnal phase and classify the timing of precipitation for each day into three categories, i.e. afternoon-to-evening precipitation (AEP), early morning precipitation (EMP), and late morning precipitation (LMP) with peaks that occur in the time windows of 1300–2400 LT, 0100–0400 LT, and 0500–1200 LT, respectively. In terms of frequency of occurrence, AEP, EMP, and LMP constitute 55 %, 18.9 %, and 26.1 % of total samples of precipitation events. In spite of the smallest percentage, EMP events are characterised by seaward (as well as landward) propagation, flat phase distribution, and large mean amplitudes. The propagating characteristics of EMP are more prominent, with indications of stronger connectivity between precipitation systems over land and ocean, when data are composited by taking the 99th percentile values in each grid to represent extreme precipitation events. The flat phase distribution of EMP events suggests that the timing of coastal precipitation is not necessarily locked to the phase of land/sea-breezes, thus, allowing precipitation to occur more randomly. Furthermore, the role of South China Sea Cold Tongue (SCS-CT) and Cross Equatorial Northerly Surge (CENS) as influencing factors for the occurrences of EMP event have also been investigated. In agreement with previous studies, we confirmed that the SCS-CT generally prevails in January and February and morning precipitation events over the northern coast of West Java mainly occurred when there was more enhanced SST cooling in the South China Sea. Additionally, we found that CENS is the most differential factor with regard to the phase of coastal precipitation. In this case, CENS is positively correlated with SCS-CT and when associated with EMP events, concurrent enhancement of CENS and SCS-CT is connected to a narrow channelling of strong surface northerly wind anomalies just offshore the Indochina and Malay Peninsula.

## 1 Introduction

Modelling the diurnal cycle of precipitation over the Maritime Continent (MC) is still one of the biggest challenges in Tropical Meteorology today because of significant biases in simulated rainfall (Love et al., 2011; Teo et al., 2011; Li et al., 2017). While models can capture the gross feature, they may overestimate or underestimate the diurnal cycle amplitudes in the simulation results, especially near coastal regions (Trilaksono et al., 2011; Love et al., 2011; Li et al., 2017). Simulation results are also found to have problematic phase errors compared to satellite rainfall data (Love et al., 2011), with a bias as large as 2−4 hours over both land and ocean in the MC (Li et al., 2017). The bias in rainfall simulation is even higher, up to 12 hours over the northern coast of West Java and varied from 3 to 15 hours over the west coast of Sumatra Island, compared to the Tropical Rainfall Measuring Mission (TRMM) satellite dataset (as interpreted from Fig. 5 in Im and Eltahir, 2017). Such deficiencies





in the models cannot be remedied by simply increasing spatial resolution (e.g., Im and Eltahir, 2017; Li et al., 2017), which indicate that models still lack key capabilities to capture fundamental mechanisms involved in the diurnal convective activities over land and ocean.

It is well known that there are contrasting temporal patterns of diurnal convective activities in the MC (e.g., Nitta and Sekine,
1994; Yang and Slingo, 2001), which result in predominantly afternoon and morning precipitation peaks respective to land and ocean (Mori et al., 2004; Qian, 2008; Qian et al., 2010; Biasutti et al., 2012; Yamanaka, 2016; Yamanaka et al., 2018). Following Houze et al. (1981), many investigators believe that this canonical pattern of diurnal convection in the MC is driven by sea-land breezes interaction with the prevailing background monsoon flows. Observations of land-sea breezes over the MC have been reported in previous studies (Hadi et al., 2000; Hadi et al., 2002; Araki et al., 2006) and the land-sea breeze
explanation for the diurnal convection seems to be consistent with the climatological study by, among others, Yang and Smith (2006) that shows relatively constant phases of the diurnal convection throughout the year, although amplitudes may vary seasonally.

On the other hand, Kikuchi and Wang (2008) pointed out that phase propagation of diurnal precipitation is prominent along the land-sea boundaries, which characterises the coastal regime of the MC and other regions of the world. In an earlier study,
Mori et al. (2004) reported that precipitating systems may propagate seaward or landward from coastal region in Sumatra Island. A more recent study by Trismidianto et al. (2016) indicates that convection over coastal region in Sumatra Island can also be induced by a decaying oceanic convective system. From results of a numerical study, Li et al. (2017) show that offshore or onshore rainfall propagation may occur over coastal region of the MC all around year. These studies imply that rainfall over coastal regions may have peaks at different times in the day depending on the initial location, and the direction and phase speed
of the propagating convective systems.

Apart from the land-sea contrast of afternoon-morning precipitation peaks, diurnal precipitation analysed from TRMM data by Im and Eltahir (2017) seems to indicate that maximum precipitation over the coastal region of West Java occurs frequently in the early morning around 0100 LT during the wet season in December–January–February (DJF) period. Additionally,
diurnal variations of precipitating clouds analysed from Boundary Layer Radar (BLR) data observed during DJF also indicate that there is a slight increase (decrease) in the frequency of occurrences of deep convective (stratiform) clouds between 0300 LT and 0600 LT (as in Fig. 9 of Renggono et al., 2001) at Serpong, which is situated in the coastal plain of West Java. These previous studies did not seem to pay much attention to the peculiar timing of precipitation over coastal region but, based on results of numerical simulation, Koseki et al. (2012) mentioned that the South China Sea Cold Tongue (SCS-CT) phenomenon
might affect the morning precipitation over northern coast of Java Island by shifting the offshore convection system to move closer to the coastal area. In a case study, maximum rainfall at 0400 LT observed during 16-25 January 2018 over northern West Java was due to a Cross Equatorial Northerly Surge (CENS) activity that coincided with SCS-CT event (Mori at al., 2018). Thus, early morning precipitation peaks seem to have strong relationship with propagating convective systems over coastal region under the influence of certain background synoptic condition. Li et al. (2017) suggested that background winds
were the most important factor for the offshore and onshore propagation of precipitation from coastal region but there are other possible mechanisms as proposed by Mori et al. (2004). Hence, phenomena related to early morning precipitation peaks and possible influencing factors like the occurrences of CENS and SCS-CT need to be more thoroughly studied to improve our understanding of the diurnal convection in the MC.

In the present study, we are particularly interested in analysing the climatology of early morning precipitation peaks over the northern coast of West Java. Not only because documentation regarding this phenomenon is still limited but also because



extreme precipitations occurring in the early morning time are potentially linked to severe flooding in the Jakarta area (Wu et al., 2007, 2013; Trilaksono et al., 2011, 2012; Sulistyowati et al., 2014; Nuryanto et al., 2019). Our basic approach is to analyse the prevalence of early morning precipitation peaks over the area of interest by classifying diurnal precipitation into several dominant patterns with regard to the phase or time of occurrences. We also analyse statistical correlations between the early
morning precipitation and predominant background conditions, especially those related to CENS and SCS-CT, and attempt to interpret the results in conjunction with propagating characteristics of convective systems over coastal region.

In the next section of this paper, we describe detailed methods to identify the occurrences of early morning precipitation peak and their possible relationship with CENS and SCS-CT events, while the results are discussed in Section 3. The last section
presents the summary of our important results and findings.

## 2 Data and Method

In order to investigate diurnal cycles of precipitation over the northern coast of West Java, we use TRMM Real-Time Multi-satellite Precipitation Analyses (MPA-RT) product of 3B41RT dataset, so-called TRMM MPA-RT dataset with a spatial resolution of 0.25° × 0.25° at hourly time interval. The TRMM MPA-RT dataset of version 6 (2000-2010) and version 7 (2011-
2016) (Huffman et al., 2007, 2016; Yong et al., 2015) contain precipitation estimates calibrated by the TRMM Combined Instrument and TRMM Microwave products (Harris et al., 2007; Liu et al., 2008; Yong et al., 2015). These data are archived, being updated in near (pseudo) real-time with a latency of 6 to 10 hours, and free to download from NASA's website at https://pmm.nasa.gov/data-access/downloads. In this case, we have obtained 16-year data record for the November through March (NDJFM) periods of November 2000 to March 2016. Other data involved in our analyses are 6-hourly wind data with
spatial resolution of 2.5° × 2.5° derived from National Center for Environmental Prediction/National Center for Atmospheric Research (NCEP/NCAR) reanalysis (Kalnay et al., 1996) and Optimal Interpolated Sea Surface Temperature (OISST) from Simple Ocean Data Assimilation (SODA) dataset (Carton and Giese, 2008) that has spatial resolution of 0.25° × 0.25° at daily time interval.

It should be noted that the TRMM MPA-RT dataset was intended to improve applications in flood prediction, with a downside
of producing more false alarms in peak flow (Harris et al., 2007). However, as it has been demonstrated in many previous works (e.g., Mori et al, 2004; Kikuchi and Wang, 2008), consistent patterns of diurnal cycle in precipitation should still be possible to be extracted from the satellite data by averaging over a relatively long period of time. Moreover, Liu et al. (2008) have used the TRMM MPA-RT dataset to analyse propagating rainfall episodes in the Bay of Bengal in May to September periods during 2002 to 2004.

In this study, we first analysed the composite diurnal patterns from the time series of gridded hourly precipitation averaged over the coastal area defined as red-line bordered polygon in Fig. 1 for the months of November through March. Additionally, we also analysed the diurnal composite of longitudinally averaged precipitation over the area defined by blue-dashed rectangle in the Fig. 1. Results of these analyses are then used to draw charts shown in Fig. 2 and will be discussed in Section 3.

In order to classify the phase or timing of precipitation, we picked samples of 24-hour data (area-averaged over red-line
bordered polygon in the Fig. 1). We then decomposed each sample into its harmonic constituents (similar method to that of Nitta and Sekine, 1994) by fitting the data (after subtracted by the daily mean) using a sinusoidal function:

$$y_k(t) = C_k cos\left(\frac{2\pi k}{T}t - \theta_k\right) \qquad (1)$$

where $y_k$ is the precipitation anomaly contributed by the $k$-th harmonic component, $C_k$ is amplitude (in mm h$^{-1}$), $t$ is time in hour, $T$ is the data period (in this case, 24 h), and $\theta_k$ (in radian) is the phase lag. It should be noted that, for each sample, the
24-hour cycle starts from 1300 LT and ends at 1200 LT on the following day by assumption that precipitation is generally



minimum during those hours. By obtaining the phase angle of the first harmonic constituent $\theta_1$, the timing of diurnal precipitation can be defined by an integer $t = n$ that maximises $y_1$ in (1) and satisfies,

$$\theta_1 \approx \frac{2\pi(t=n)}{24}, n = 0,..,23 \qquad (2)$$

Note that precipitation data are given as discrete time series at hourly time interval but $\theta_1$ is computed as real number. The

classification of phase or timing of precipitation is made in two steps. In the first step, the timing of precipitation is classified into:

i.   Afternoon-to-evening precipitation (AEP) that occurs between 1300 and 2400 LT, or $0 \leq \theta_1 \leq (\frac{11}{12})\pi$.

ii.  Night time-to-morning precipitation (NMP) that occurs between 0100 and 1200 LT on the following day, or $\pi < \theta_1 < (\frac{23}{12})\pi$.

Then, in the second step, samples belonging to NMP are further classified into:

i.   Early morning precipitation (EMP) that occurs between 0100 and 0400 LT (on the following day), or $\pi < \theta_1 \leq (\frac{15}{12})\pi$.

ii.  Late morning precipitation (LMP) that occurs between 0500 and 1200 LT, or $(\frac{16}{12})\pi < \theta_1 < (\frac{23}{12})\pi$.

The first classification is performed in order to check whether early morning precipitation events are distinguishable from other samples, whereas the second classification is to quantify the prevalence of early morning precipitation relative to all

samples in the NMP category.

Furthermore, to investigate factors that influence early morning (0100–0400 LT) precipitation in the northern coast of West Java, we carried out analyses of meridional wind and SST fields attributive to CENS and SCS-CT phenomena. In this case, we followed Hattori et al. (2011) and Koseki et al. (2012) to calculate indices for CENS and SCS-CT respectively. Details of the calculations can be briefly described as follows:

a.   The CENS index is calculated from daily mean surface (925 hPa) level meridional wind data spatially averaged over the area of $(0{-}5^o\,S, 105{-}115^o\,E)$ as depicted in Fig. 8. Considering that the lifetime of a cold surge event is around 5–14 days with an average duration of approximately 7 consecutive days (Zhang et al., 1997), the resulted daily time series of wind data are then smoothed by 5-day moving average. The smoothed spatially-averaged time series data of the meridional wind thus defines the daily CENS index, and a threshold of $-4.5\ m^{-2}$, is used to determine the occurrence of

CENS.

b.   Similar method applies for obtaining the SCS-CT index, but with the SST data averaged over the area of $(2{-}10^o\,N, 105{-}112^o\,E)$ as shown in Fig. 9. In this case, a day with SCS-CT event is defined when the index is lower than the threshold temperature of $26.4^o\,C$ (see Koseki et al., 2012, for more detailed procedures and explanations).

c.   The CENS and SCS-CT indices are then used to calculate the relative frequency of occurrences of the coincidence

30       between CENS and/or SCS-CT events and the three phases of precipitation peaks determined from previously mentioned harmonic analysis.

As we determined the phase of precipitation peaks based only on the first harmonic constituent that has the largest amplitude, we have one uniquely classified sample for each day. The samples of dates collected in these analyses were then used to make various composite plots.

**3 Results and Discussion**

**3.1 Climatology of Early Morning Precipitation Peaks**

The diurnal composites of area averaged precipitation over the northern coast of West Java (red-line bordered polygon in Fig. 1, for the months of November to March during 2000 to 2016, are depicted in Fig. 2a. It can be seen that the dominant pattern is characterised by precipitation peaks occurring in the afternoon to evening time (1500–2100 LT). However, diurnal



precipitation patterns in the months of January and February exhibit significant secondary precipitation peaks in early morning to morning time (0100–0700 LT). The Hovmöller diagrams in Fig. 2b-f show the time-latitude variations of longitudinally averaged rainfall over the blue-dashed rectangle in Fig. 1. It should be clear that the coastal region (marked by vertical black dashed-lines) is affected by precipitation systems from both inland and oceanic origins. Figures 2d and 2e also clearly indicate

that the oceanic precipitation system penetrates further inland during January and February, which appear as secondary peaks occurring between 0100 and 0700 LT as shown in Fig. 2a. In addition to two stationary patterns of precipitation over land and sea, Figures 2c-f also show weak signals of land-to-sea propagation of precipitation over coastal region during the night-to-morning transition between 2300 LT and 0300 LT on the following day.

The timing of precipitation peaks over the coastal region has been further analysed from the results of harmonic analysis

(described in previous section). Herein, we focus on the data of January and February that show strong signals of morning precipitation peaks (see Fig. 1). The left panels of Figure 3 show the first harmonic (diurnal) constituent of each sample of 24-hour rainfall time series averaged over the coastal region defined in Fig. 1. The upper-left panel of Figure 3 shows the results for data that are categorised as AEP with maximum values between 1300 and 2400 LT, while Fig. 3b shows data of NMP category with maximum values between 0100 and 1200 LT. Figures 3a and 3b clearly demonstrate that the application of the

harmonic analysis method was successful to classify the timing of diurnal precipitation. It is found that AEP is a dominant feature of the diurnal precipitation, which contributes 55% of the diurnal rainfall patterns over the studied area. However, NMP also has high frequency of occurrence contributing 45 % of total samples in January and February throughout 2001 to 2016 period.

In Fig. 3c, three composites of the first harmonic signals are depicted. The first composite (red curve) represent the mean

sinusoidal signal of AEP, while that of NMP has been further split into EMP that occurs between 0100 and 0400 LT (green curve) and LMP whose time of occurrence is between 0500 and 1200 LT. The frequency of occurrence of EMP and LMP are 42 % and 58 % respectively, relative to total NMP samples, or 18.9 % and 26.1 % relative to all samples. It is of interest to note that, in spite of the smallest percentage, EMP has the largest mean amplitude.

Figure 4 shows more detailed analysis of the phase distribution and the mean amplitude for all of the three 24-hour cycle

precipitation patterns, based on the results of harmonic analysis. It can be seen that phases of AEP exhibits a nearly normal distribution centred around 1900 LT with two maxima of mean amplitudes around 1800 LT and 2200 LT. On the other hand, LMP show a distribution that decreases with time resembling a gamma-like distribution with maximum mean amplitude around 0600 LT. These characteristics of the distribution of frequency of occurrence of the two types of precipitation are indicative of the concentration times of convective activities, which is around 1900 LT over land and 0600 LT over sea. These results are in

agreement with many previous investigations (e.g., Qian, 2008) and seem to signify the canonical pattern of land-sea contrast in the phase of precipitation, which are believed to be mainly controlled by land–sea breezes and their interactions with the monsoonal flows (e.g., Houze et al., 1981). However, it is quite interesting that EMP events have not only a flat phase distribution, which is indicating more random events, but also high mean amplitudes with peaks around 0400 LT. The random occurrence of early morning precipitation might be closely related to propagating systems, as indicated from Fig. 2b-f, since

their phases are not necessarily locked to the timing of sea/land-breezes.

### 3.2 Relationships with Propagating Convective Systems and Extreme Precipitation Events

In order to investigate the relationship between early morning precipitation and the propagating characteristics of precipitation system over coastal region, we constructed Hovmöller diagrams similar to that of Fig. 2b-f but with data classified according to the previously defined three types of precipitation peaks i.e. AEP, EMP, and LMP and show the results in Fig. 5. Propagation

of precipitation connecting two relatively stationary systems over land and sea can be distinctly seen from Fig. 5a and 5b. The





propagating precipitation can be associated with rainfall events that occur during late night (Fig. 5a) and early morning (Fig. 5b) time over coastal region. It should be noted that in Fig. 5a the direction is mainly one way from land to sea, while in Figure 5b there is also discernible land-ward pattern of propagation. On the other hand, Figure 5c illustrates more detached precipitation systems over land and sea. In this case, late morning precipitation over the coastal region can be seen as an

extension of the oceanic precipitation system.

We have discussed earlier in Section 1 that EMP may be linked to extreme precipitation events over the northern coast of Java Island. We performed other analysis using Hovmöller diagrams, similar to those in Fig. 5, but with averaged precipitations are of the 99$^{th}$ percentile (P99) to represent extreme values in each data grid. The results, which are depicted in Fig. 6, strongly suggest that extreme rainfall events over the coastal region mainly occur between 0100 and 0400 LT and characterised by

propagating systems (Fig. 6b). In this case, although oceanic convection seems to be major determinant, linkage to land-based precipitation may also be important. It should also be clear that extreme precipitation events that occurred during late-night (Fig. 6a) and late-morning (Fig. 6c) time have single origin of either land-based or oceanic convection.

### 3.3 Relationships with CENS and SCS-CT

As discussed in previous Section, we have objectively identified EMP events over the northern coast of West Java from satellite
data using harmonic analysis. We have also found that EMP is strongly associated with propagating precipitation systems that show connection between land-based and oceanic convections. While we have insufficient data to analyse the propagating precipitation systems in more detail, synoptic conditions favourable for their occurrence may be investigated from global reanalysis data. We also mentioned earlier that such favourable condition may develop under the influence of CENS, or SCS-CT, or the combination of both.

We analysed the frequency of occurrence of CENS and SCS-CT events corresponding to each classification of diurnal precipitation patterns of AEP, EMP, and LMP. Figure 7a shows frequency of occurrence of CENS, SCS-CT, and combined events, which are calculated relative to the number of samples in each diurnal precipitation namely: 205 for AEP, 51 for EMP, and 119 for LMP. It should be noted that the frequencies do not sum up to 100 % because some samples are considered unclassified if they are not associated with either CENS or SCS-CT, and there are also overlapping number of samples between
CENS and SCS-CT events. Bearing in mind that we focus this analysis to data of January and February, it can be seen from Figure 7 that SCS-CT generally prevails with frequency of occurrence above 50 % in all classified events. However, SCS-CT exhibits stronger association with morning precipitation events with frequency of occurrence above 70 % in both EMP and LMP categories. This seems to indicate possible influence of SCS-CT that may induce land-ward shifting of precipitation system over the Java Sea (Koseki et al., 2012).

On the other hand, the frequency of occurrence of CENS is the highest for samples belonging to EMP and almost negligible for AEP category. It should be noted that the number of CENS events are almost the same for both EMP and LMP categories and, although we uniquely classify the timing of diurnal precipitation using harmonic analysis, some of the events may be interrelated due to subsequent processes from the previous or in the following day. As it is evidenced in Fig. 7a and depicted more clearly in Fig. 7b, most CENS events are in coincidence with SCS-CT and their magnitudes are positively correlated.

The composite maps of SST and meridional wind fields, corresponding to SCS-CT and CENS those are classified according to the phase of precipitation, are shown in Fig. 8 and 9. In Figures 8b and 8c, contours of the negative SST anomalies (after subtracted by the threshold value of 26.4° C) protrude relatively closer towards the equator that indicates the strengthening of the SCS-CT when associated with morning precipitations over the northern coast of West Java. Although differences in the SST fields for EMP and LMP events are not quite discernible, the composite maps of wind fields and CENS index in Figure 8



show significantly different patterns. The enhancement of CENS associated with EMP events (Fig. 8b) is connected to an elongated channelling of strong northerly wind anomalies offshore the peninsular of Indochina, Malay, down to Karimata Strait; the northerly winds are even observed to cross Java Island. On the other hand, LMP events (Fig. 8c) are characterised by an enhancement of the northerly winds at larger spatial scale and stronger connection with extratropical regions, but weaker

magnitudes over the Java Sea compared to that of EMP case.

By this far, our results indicate and confirm previous works (Wu et al., 2007; Hattori, 2011; Mori et al. 2018) that CENS is an important factor for the occurrence of EMP events over the northern coast of West Java and cooler SST in the South China Sea seems to provide a favourable background condition. In addition, we found that the enhancement of northerly winds during the EMP events may have more localised origin over the South China Sea. The coincidence of SCS-CT and CENS events

brings forward the aspect of mesoscale sea-air interaction with its all possible feedbacks, which is quite interesting because cold surge that reaches the tropics is traditionally viewed as the effects of midlatitude weather during boreal winter (e.g., Chang et al., 2005); but more works are needed to make further analysis.

## 4 Summary and Conclusion

In the present work, we have studied the diurnal cycles of precipitation over the northern coast of West Java, with emphasis

on the occurrences of early morning precipitation peaks, mainly using the TRMM MPA-RT dataset spanning from 2000 to 2016. The diurnal composites of precipitation averaged over the study area for the months of November to March are dominantly characterised by precipitation peaks that occur in the afternoon to evening time (1500–2100 LT). However, diurnal precipitation patterns in January and February show secondary precipitation peaks in night to morning time (0100–0700 LT). Results of Hovmöller analysis revealed that, beside the canonical pattern of afternoon (morning) convections over land (sea),

there are weak signals of land-to-ocean propagation during the night-to-morning transition time. This study was further motivated by this propagating characteristic of precipitation over coastal regions in the MC that have been suggested by previous studies (Mori et al., 2004; Kikuchi and Wang, 2008; Li et al., 2017).

We then applied harmonic analysis method to objectively determine the phase of precipitation peaks over the northern coast of West Java focusing on data of January and February, whereby the timing of precipitation for each day has been classified

into three categories: (1) afternoon-to-evening precipitation (AEP) that occurs between 1300 and 2400 LT, (2) early morning precipitation (EMP) that occurs between 0100 and 0400 LT, and (3) late morning precipitation (LMP) that occurs between 0500 and 1200 LT. The EMP contributes 18. 9 % of total samples analysed, which is a minority compared to that of the LMP (26.1 %), and AEP (55 %). However, we found that the EMP events are characterised by: (1) sea-ward as well as land-ward propagation with indication of stronger connectivity between land-based and oceanic precipitation systems, (2) flat phase

distribution, and (3) high mean amplitudes peaking around 0400 LT. These propagating characteristics of EMP are more prominent when data are composited by taking the 99[th] percentile (P99), which signifies the importance of EMP in association with extreme precipitation events. The flat phase distribution of EMP events suggests that the timing of coastal precipitation is not necessarily locked to the phase of land/sea-breezes, thus, allowing precipitation to occur more randomly. This implies that, even for as short as 24-hour lead time, probabilistic forecast maybe necessary to assess the hazard of heavy precipitation

in this region.

Following the lead of previous studies, we investigated the role of SCS-CT and CENS as influencing factors for the occurrences of EMP event. By calculating the frequency of occurrence of SCS-CT and CENS that correspond to the three timing of the costal precipitation, it is found that SCS-CT generally prevails in January and February but morning precipitation events over the northern coast of West Java mainly occurred in coincidence with enhanced SST cooling in the South China Sea, which is



in agreement with the numerical modelling results of Koseki et al. (2012). We also found that CENS is the most important factor that influences the occurrence of EMP and its association with extreme precipitation events over the northern coast of West Java, which is consistent with results from previous studies (Wu et al., 2007; Hattori, 2011; Mori et al. 2018). Most of the CENS events identified in this study occurred in coincidence with SCS-CT and both indices exhibit positive correlation

when associated with EMP. In this case, the enhancement of CENS and SCS-CT is connected to an elongated channelling of strong northerly wind anomalies just offshore the Indochina, Malay Peninsula, and even observed to cross the Java Island.

In the present study, we are aware of the issues concerning TRMM MPA-RT dataset to represent diurnal precipitation in the MC but we have presented results that are consistent with previous studies and produced some new documentations about the propagating precipitation systems over coastal region. Similar or extension to this study will take more benefit from better

satellite rainfall products. We have not yet investigated the mechanisms responsible for the propagation of the coastal precipitation but the role of cold pool (e.g., Teo et al., 2011; Trismidianto et al., 2016) may serve as a concrete and testable hypothesis to be the manifestation of the so-called self-replicating mechanism proposed by Mori et al. (2004). We additionally found that the CENS associated with EMP may have more localised origin and seems to involve mesoscale air-sea interactions over the South China Sea, which is of interest to address in our future study.

**Acknowledgments**

The authors are thankful to the Indonesia Endowment Fund for Education (LPDP) as a main sponsor of this research. The co-authors also partially supported by ITB P3MI Research Grants 2017–2019. The authors would also like to express their appreciation to Prof. Shigeo Yoden of Kyoto University and other research team members under the JSPS and DG-RSTHE Joint Research Program for FY 2018–20, for their useful discussions and suggestions.

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

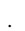

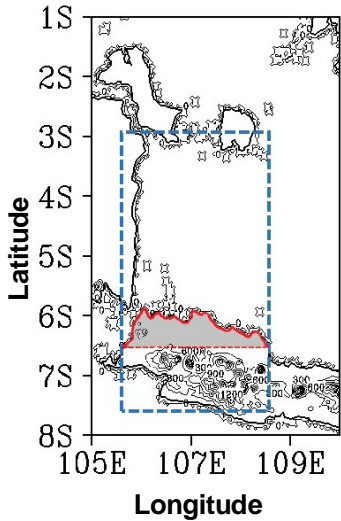

**Figure 1. A map showing the main area of interest in this study; with coastlines and topographic contours. Red-line bordered polygon (with land masks over the northern coastal plain of West Java) defines the area for spatial averaging of gridded precipitation data. Blue-dashed rectangle defines the area for longitudinally ($105.5 - 108.5°$ E) averaged precipitation in Hovmöller analysis of Fig. 2.**




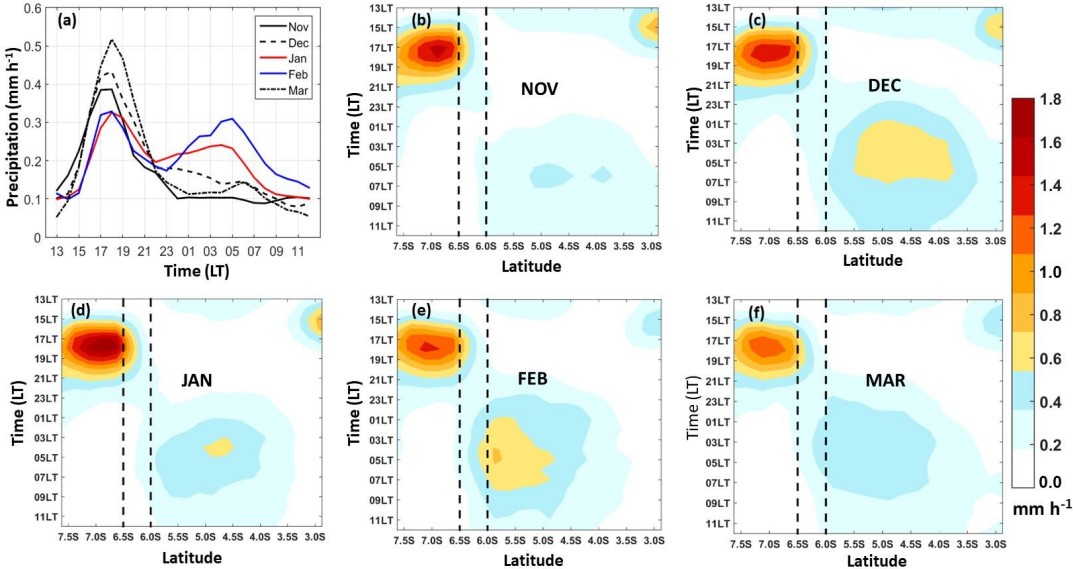

**Figure 2. (a)** Climatology of area-averaged (see Figure 1) diurnal precipitation over the northern coast of West Java, depicted as line plots of 24-hour composite time series in different colours for the months of November (solid black), December (dashed black), January (solid red), February (solid blue), and March (dashed-dotted black) analysed from TRMM MPA-RT dataset of 2000–2016 period. Other panels show the corresponding time-latitude cross sections (Hovmöller diagrams) of the diurnal composites for the months of: **(b)** November; **(c)** December; **(d)** January; **(e)** February; and **(f)** March. Dashed vertical black lines in Fig. 2(b-f) denote the latitudes of northern coastal area of West Java. Averaged precipitation rates in mm h$^{-1}$ are shown as shaded contours.

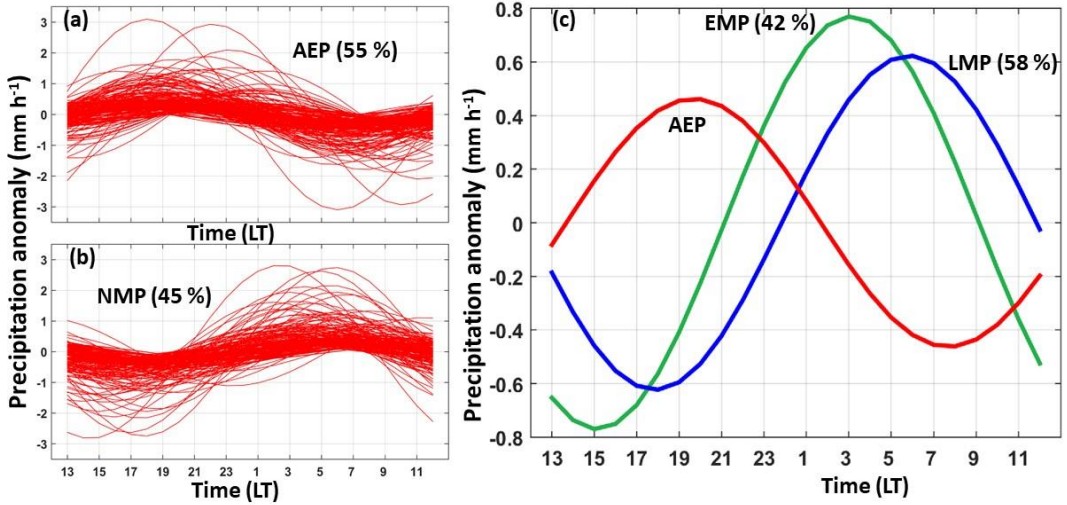

**Figure 3.** The first harmonic components analysed from diurnal precipitation data that are grouped into: **(a)** Afternoon-to-evening precipitation (AEP) between 1300-2400 LT, and **(b)** Night time-to-morning precipitation (NMP) between 0100-1200 LT based on the phase angles (see text for explanation); **(c)** composite sinusoidal curves of AEP (red line) with those of NMP split into: early morning precipitation (EMP) between 0100-0400 LT and late morning precipitation (LMP) between 0500-1200 LT. Percentage indicates frequency of occurrence relative to total samples for AEP and NMP, but those of EMP and LMP are relative to NMP samples.



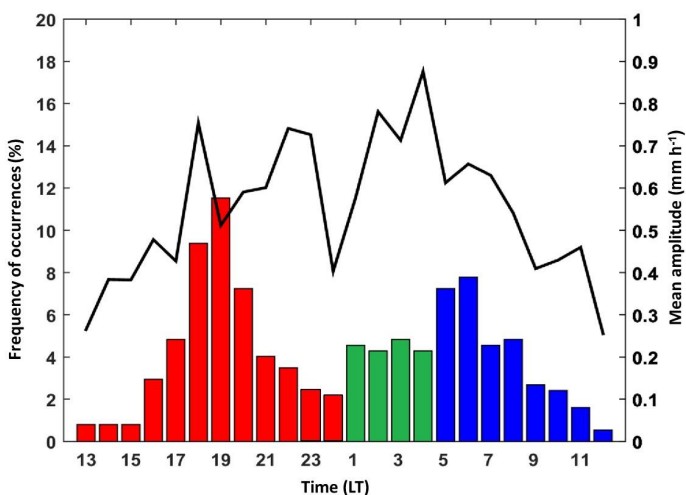

**Figure 4. Diurnal distribution of the mean amplitude of the first harmonic components (solid black line) and frequency of occurrence of the peak precipitation time (coloured bar chart) differentiated into three groups (same as in Figure 3): AEP (red), EMP (green), and LMP (blue).**

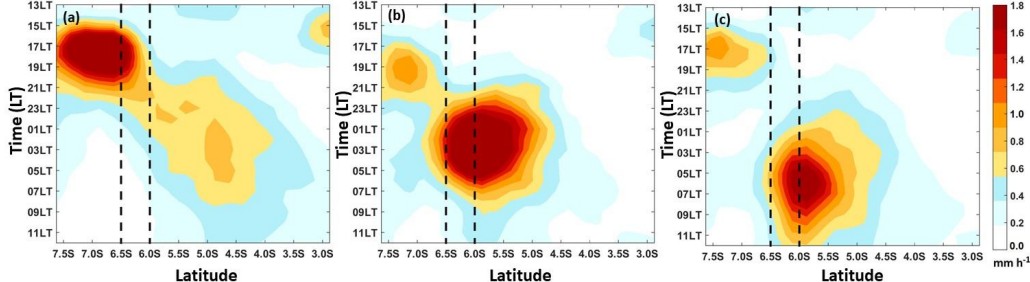

**Figure 5. Hovmöller diagrams of composite diurnal precipitation; similar to Figure 2 except data are classified by the phase or the timing of peak precipitation (as in Fig. 3): (a) AEP, (b) EMP, and (c) LMP.**

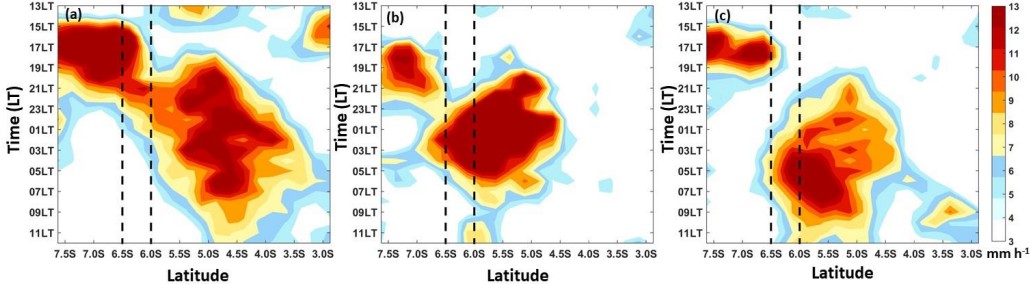

**Figure 6. Hovmöller diagrams, same as Fig. 5, except averaged data are the 99th percentile (P99) values of the gridded hourly precipitation.**





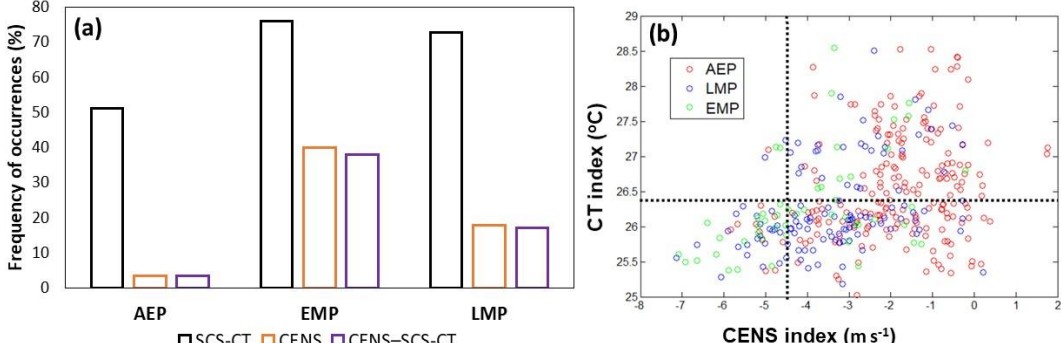

**Figure 7. (a) Frequency of occurrence of SCS-CT (black), CENS (orange), and CENS–SCS-CT (purple) corresponding the three diurnal phases of peak precipitation: AEP, EMP, and LMP; (b) scatter plot between SCS-CT and CENS indices for: AEP, LMP, and EMP as indicated by red, blue, and green circles respectively. Thresholds values for SCS-CT ($26.4°$ C) and CENS ($-4.5$ m s$^{-1}$ ) indices are indicated by black dotted lines.**

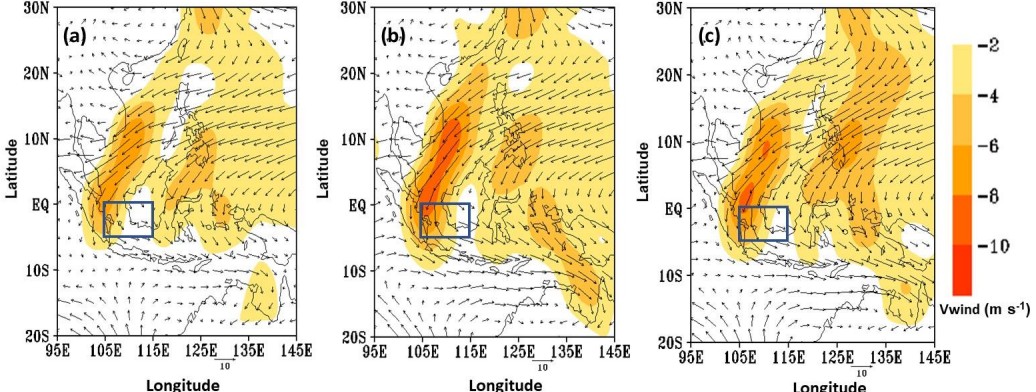

**Figure 8. Spatial distribution of mean (composite) 850 hPa horizontal wind vectors and magnitude of meridional wind (shaded) corresponding to the three diurnal phases of preak precipitation: (a) AEP, (b) EMP, and (c) LMP. The dark-blue solid rectangle represents the CENS index area (following Hattori et al., 2011).**

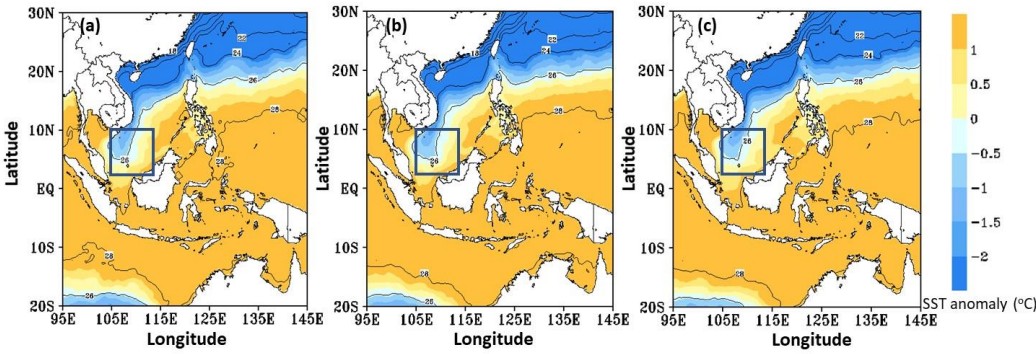

**Figure 9. Same as Figure 8, but for SST (solid line contour) and SST anomaly relative to SCS-CT threshold of $26.4°$ C (shaded). The red solid rectangle represents the SCS-CT index area (following Koseki et al., 2012).**