# Peer review of "Early Morning Peaks in the Diurnal Cycle of Precipitation over the Northern Coast of West Java and Possible Influencing Factors"

_Annales Geophysicae, 2019_

## Referee Comment (RC1) · Anonymous Referee #1 · 30 Aug 2019

This manuscript investigated the diurnal cycle of precipitation over the northern coast of West Java with a focus on early morning precipitation and influence from SCS-CT and CENS. Well-chosen classification method has clarified the seasonally changing diurnal cycle pattern. Strong correlation between CENS and extreme EMP is also clarified, however, there is no clear link between SCS-CT and variation of diurnal cycle patterns. Therefore, the reviewer would suggest accepting the manuscript for publication after minor revisions.

P1, L20: "characterised by seaward (as well as landward) propagation" As is shown in P6, L10, landward propagating oceanic convection is major determinant of EMP in the

northern coastal area, though seaward propagation in the inland area in the nighttime may have some linkage. There is a gap in this description and different from the fact in Figs 10 and 11.

P4, L24: -4.5m-2 -> m s-1

P5, L7: "Figures 2c-f also show weak signals of land-to-sea propagation of precipitation over coastal region during the night-to-morning transition between 2300LT and 0300LT on the following day." It is hard to recognize land-to-sea propagation in Figs 2e and 2f.

P5, L11: (see Fig. 1) -> Fig. 2a

P6, L36: In Figures 8b and 8c, -> In Figures 9b and 9c

P6, L37: "relatively closer towards the equator that indicates the strengthening of the SCS-CT when associated with morning precipitation over the northern coast of Java." The difference between the three figures (9a-c) is indistinguishable.

P13, Figure 9: The red solid rectangle -> The dark-blue solid rectangle

---

## Referee Comment (RC2) · Anonymous Referee #2 · 10 Nov 2019

Overview: The study combines 17 years of TRMM satellite and multi-sensor rainfall data, reanalysis datasets, and SST observations to investigate the climatology of diurnal rainfall patterns in the region of northwestern Java, the largest east–west-oriented island in the western Maritime Continent. While the focus region is highly localized, the Maritime Continent is an exceedingly complex region, thus warranting study of potentially unique dynamics in different sub-regions. Further, it is possible that the results extend to other regions in the Maritime Continent, to some measure, though further work would be necessary to determine this.

Recommendation: Accept after minor revisions. Overall, the study is clear and con-

cise, well-organized, and supported in its claims, with some minor exceptions as noted below.

Comments:

Background context and propagation mechanisms: A growing body of studies argue for the likely importance of gravity waves in governing diurnal offshore rainfall propagation, which often manifests at phase speeds faster than the nocturnal land breeze alone can explain, as first highlighted by Mapes et al. (2003). Two studies that argue for this mechanism in the Maritime Continent are Love et al. (2011) and Ruppert and Zhang (2019). It might be useful to note this mechanism since it is likely relevant to the findings.

Love, B. S., A. J. Matthews, and G. M. S. Lister, 2011: The diurnal cycle of precipitation over the Maritime Continent in a high-resolution atmospheric model. Q. J. R. Meteorol. Soc., 137, 934–947, doi:10.1002/qj.809.

Mapes, B. E., T. T. Warner, and M. Xu, 2003: Diurnal Patterns of Rainfall in Northwestern South America. Part III: Diurnal Gravity Waves and Nocturnal Convection Offshore. Mon. Weather Rev., 131, 830–844, doi:10.1175/1520-0493(2003)131<0830:DPORIN>2.0.CO;2.

Ruppert, J. H., and F. Zhang, 2019: Diurnal Forcing and Phase Locking of Gravity Waves in the Maritime Continent. J. Atmos. Sci., 76, 2815–2835, doi:10.1175/JAS-D-19-0061.1.

While propagation is clearly evident in some of the panels, text like the following may not be fully justified by the figures and results (P.6 L11–12): "It should also be clear that extreme precipitation events that occurred during late-night (Fig. 6a) and late-morning (Fig. 6c) time have *single origin* of either land-based or oceanic convection" (I placed asterisks for emphasis on what I think is dubious). Perhaps the authors would agree, that evidence of propagation does not necessarily mean that the offshore and inland

rainfall peaks would not exist without this propagation. Perhaps it is equally plausible that some mechanism(s) favor rainfall in both regions, independent of the propagation (especially when these peaks are quite separate, as in Fig. 6a)?

P. 6 L28: "land-ward shifting of precipitation" Again, Figs. 5 and 7a seem to suggest that the SCS-CT favors *offshore* rainfall. I do not understand why the emphasis is placed specifically on land-ward propagation, based on these figures alone.

Colors in Figs. 5 and 6 are saturated, making it difficult to interpret relative rainfall magnitudes.

Editorial comments:

P.4 L24: "$-4.5$ m$-2$" I think you mean m s-1?

P.5 L12–13: Just to clarify, does Fig. 13a,b show (N-days*17-years) samples, or have they been averaged by month? Please indicate in the text.

P.5 L22–23: Could this be due to sampling? I.e., EMP is greatest since it has been averaged over fewer samples than the other categories?

P.6 L36: Should be "In Figures 9b and 9c,..."

Fig. 1: Consider expanding the domain shown to provide a broader context for those less unfamiliar with this region.

---

## Author Comment (AC1) · 17 Nov 2019

General comment from Anonymous Referee #1 (hereafter Comment) #1: This manuscript investigated the diurnal cycle of precipitation over the northern coast of West Java with a focus on early morning precipitation and influence from SCS-CT and CENS. Well-chosen classification method has clarified the seasonally changing diurnal cycle pattern. Strong correlation between CENS and extreme EMP is also clarified, however, there is no clear link between SCS-CT and variation of diurnal cycle patterns. Therefore, the reviewer would suggest accepting the manuscript for publication after minor revisions.

Response by Authors (hereafter Response) #1: The authors greatly appreciated the constructive comments from the Reviewer. With regard to the link between SCS-CT and diurnal cycle patterns over the northern coast of West Java, we have actually pointed out as follows: For all months from November to March, the diurnal precipitation pattern is characterised by a main peak during evening to night time. However, in January and February prominent secondary peaks are also observed in the morning time (Figure 2a). SCS-CT generally prevails in January and February and morning precipitation events over the northern coast of West Java mainly occurred when there was more enhanced SST cooling in the South China Sea (Abstract: Lines 27-28). In this study, we further classifed morning precipitatin events into early morning and late morning phase groups. We found that the early morning precipitation events were more strongly correlated with CENS, whereas CENS is positively correlated with SCS-CT. Therefore, we can say that SCS-CT has the role to induce morning precipitation over the coastal region of West Java (in agreement with Koseki et al., 2012). SCS-CT also play a role as background condition for the occurrence of CENS that in turn induced early morning precipitation events.

Comment #2: P1, L20: "characterized by seaward (as well as landward) propagation" As is shown in P6, L10, landward propagating oceanic convection is major determinant of EMP in the northern coastal area, though seaward propagation in the inland area in the nighttime may have some linkage. There is a gap in this description and different from the fact in Figs 10 and 11.

Response #2: We thank the Reviewer for the comment regarding a gap between depiction of figures and the figures themselves. However, we would like to clarify that Figures 10 and 11 mentioned by the Reviewer are actually Figures 5 and 6. The gap as mentioned by the Reviewer seems to refer to direction of propagation in Figures 5 and 6, which also applies to Figures 2b-f. Please note that the 24-hourly precipitations data are sorted from 1300 LT to 1200 LT in the following day. Therefore, in those figures, time increases from top to bottom. In this case, seaward as well as landward propagation of precipitation systems could be misinterpreted if time direction is reversed. To minimize such misinterpretation, we have added annotation for time directions and also white lines with arrows that can guide readers to follow the direction of propagation (following Mori et al., 2004) in Figure 5 (a) - (b). Comment #3: P4, L24: -4.5m-2 -> m s-1

Response #3: We thank the Reviewer for the typo correction in P4, L24. We have already changed the unit of wind from: Previous unit: -4.5 ms^(-2) Current unit: -4.5 ms^(-1)

Comment #4: P5, L7: "Figures 2c-f also show weak signals of land-to-sea propagation of precipitation over coastal region during the night-to-morning transition between 2300LT and 0300LT on the following day." It is hard to recognize land-to-sea propagation in Figs 2e and 2f.

Response #4: The author thanks the Reviewer for the thoughtful comment. We recognize that regarding Figures 2 (e) and 2 (f), it is indeed difficult to argue that there is a weak signal of propagation between land and sea. This statement is based on a visual interpretation that precipitation intensity for larger than 0.2 mm of landward propagation of precipitation which is occurred from late afternoon to late night, and continuing from midnight to early morning in the following day. To clarify the connection between land and sea convection systems, we add contour lines to Figure 2 (b) - (f).

Comment #5: P5, L11: (see Fig. 1) -> Fig. 2a

Response #5: We thank the Reviewer for the typo correction in P5, L11. We have already changed the word from: Previous: (see Fig. 1) Current: (see Fig.2a)

Comment #6: P6, L36: In Figures 8b and 8c, -> In Figures 9b and 9c

Response #6: We thank the Reviewer for the typo correction in P6, L36. We have already changed the word from: Previous: In Figures 8b and 8c Current: In Figures 9b and 9c

Comment #7: P6, L37: "relatively closer towards the equator that indicates the strengthening of the SCS-CT when associated with morning precipitation over the northern coast of Java." The difference between the three figures (9a-c) is indistinguishable.

Response #7: The authors would like to thank the Reviewer for the constructive comment. The authors are aware that there is no significant difference between SCS-CT changes in the diurnal AEP, EMP, LMP which are exhibited in Figures 9 (a) - (c). Therefore, we consider further analysis of the variation in the spatial pattern of SCS-CT events between no CENS (NCENS) and CENS in the EMP samples. This analysis shows spatially the SCS-CT composite between NCENS and CENS which described that while CENS occurred, SCS-CT appeared wider and relatively closer to equator. Differences in spatial patterns between both of SCS-CT NCENS and CENS are depicted in Figure 10.

Comment #8: P13, Figure 9: The red solid rectangle -> The dark-blue solid rectangle.

Response #8: We thank the Revirewer for the correction of wrong word in caption of figure 9 (P13). We have already changed the word from: Previous: The red solid rectangle Current: The dark-blue solid rectangle

Please also note the supplement to this comment:
https://www.ann-geophys-discuss.net/angeo-2019-107/angeo-2019-107-AC1-supplement.pdf

---

## Author Comment (AC2) · 17 Nov 2019

We thank to Anonymous Referee #2 for the general and detail constructive comments which is submitted on 10 November 2019. However, to adequately address the concerns raised by Reviewer 2 regarding the original manuscript, authors need additional times at about one week later. Thus, we try to submit the response not later than 22 November 2019. If the interactive discussion will be closed on 17 November 2019, we will submit our Author Comments by e-mail to Editor.

---

## Author Comment (AC3) · 17 Nov 2019

Dear Reviewer 1,

The authors thank the Reviewer 1 for the insightful comments. To adequately address the concerns raised by Reviewer 1 regarding the original manuscript, we have made the following changes in figures as attached below. We have also already submitted the detail explanation in separate reply about Author Response for Anonymous Referee #1.

Figure 2. (a) Climatology of area-averaged (see Figure 1) diurnal precipitation over the

northern coast of West Java, depicted as line plots of 24-hour composite time series in different colours for the months of November (solid black), December (dashed black), January (solid red), February (solid blue), and March (dashed-dotted black) analysed from TRMM MPA-RT dataset of 2000–2016 period. Other panels show the corresponding time-latitude cross sections (Hovmöller diagrams) of the diurnal composites for the months of: (b) November; (c) December; (d) January; (e) February; and (f) March. Dashed vertical black lines in Fig. 2(b-f) denote the latitudes of northern coastal area of West Java. Averaged precipitation rates in mm h-1 are shown as shaded contours.

Figure 5. Hovmöller diagrams of composite diurnal precipitation; similar to Figure 2 except data are classified by the phase or the timing of peak precipitation (as in Fig. 3): (a) AEP, (b) EMP, and (c) LMP. The black dashed line indicates boundary of coastal area. The white solid arrow represents direction of propagation (following Mori et al., 2004), with seaward propagation is symbolized asïÅăïÅíïĄąïÅľïÅăand landward propagation is written asïÅăïÅíïĄćïÅľïÅő

Figure 6. Hovmöller diagrams, same as Fig. 5, except averaged data are the 99th percentile (P99) values of the gridded hourly precipitation.

Figure 9. Same as Figure 8, but for SST (black solid line contour) and SST anomaly relative to SCS-CT threshold of ãĂŰ26.4ãĂŮˆo C (shaded). The dark-blue solid rectangle represents the SCS-CT index area (following Koseki et al., 2012).

Figure 10. Same as Figure 9, but for composite of EMP samples: a) SCS-CT without CENS, b) SCS-CT with CENS.
* * *
[Figure]

**Fig. 1.** Figure 2. (a) Climatology of area-averaged (see Figure 1) diurnal precipitation over the northern coast of West Java, depicted as line plots of 24-hour composite time series in different colours for t

[Figure]

**Fig. 2.** Figure 5. Hovmöller diagrams of composite diurnal precipitation; similar to Figure 2 except data are classified by the phase or the timing of peak precipitation (as in Fig. 3): (a) AEP, (b) EMP, and (

[Figure]

**Fig. 3.** Figure 6. Hovmöller diagrams, same as Fig. 5, except averaged data are the 99th percentile (P99) values of the gridded hourly precipitation.

[Figure]

**Fig. 4.** Figure 9. Same as Figure 8, but for SST (black solid line contour) and SST anomaly relative to SCS-CT threshold of ãĂŰ26.4ãĂŮˆo C (shaded). The dark-blue solid rectangle represents the SCS-CT index area

[Figure]

**Fig. 5.** Figure 10. Same as Figure 9, but for composite of EMP samples: a) SCS-CT without CENS, b) SCS-CT with CENS.

---

## Author Comment (AC4) · 26 Nov 2019

General comment from Anonymous Referee #2 (hereafter Comment) #1: The study combines 17 years of TRMM satellite and multi-sensor rainfall data, reanalysis datasets, and SST observations to investigate the climatology of diurnal rainfall patterns in the region of northwestern Java, the largest east–west-oriented island in the western Maritime Continent. While the focus region is highly localized, the Maritime Continent is an exceedingly complex region, thus warranting study of potentially unique dynamics in different sub-regions. Further, it is possible that the results extend to other regions in the Maritime Continent, to some measure, though further work would be

necessary to determine this.

Response by Authors (hereafter Response) #1: We greatly thank the Reviewer for giving us the insightful comments. We really appreciated the Reviewer's recogniton of our small contribution and its potential extention to further works that will hopefully lead to better understanding of the building blocks of weather dynamics in the Maritime Continent.

Comment #2: Background context and propagation mechanisms: A growing body of studies argue for the likely importance of gravity waves in governing diurnal offshore rainfall propagation, which often manifests at phase speeds faster than the nocturnal land breeze alone can explain, as first highlighted by Mapes et al. (2003). Two studies that argue for this mechanism in the Maritime Continent are Love et al. (2011) and Ruppert and Zhang (2019). It might be useful to note this mechanism since it is likely relevant to the findings.

Love, B.S., A.J. Matthews, and G.M.S.Lister, 2011: The diurnal cycle of precipitation over the Maritime Continent in a high-resolution atmospheric model. Q. J. R. Meteorol. Soc., 137, 934–947, doi:10.1002/qj.809.

Mapes, B. E., T. T. Warner, and M. Xu, 2003: Diurnal Patterns of Rainfall in Northwestern South America. Part III: Diurnal Gravity Waves and Nocturnal Convection Offshore. Mon. Weather Rev., 131, 830–844, doi:10.1175/15200493(2003)131<0830:DPORIN>2.0.CO;2.

Ruppert, J. H., and F. Zhang, 2019: Diurnal Forcing and Phase Locking of Gravity Waves in the Maritime Continent. J. Atmos. Sci., 76, 2815–2835, doi:10.1175/JAS-D19-0061.1.

While propagation is clearly evident in some of the panels, text like the following may not be fully justified by the figures and results (P.6 L11–12): "It should also be clear that extreme precipitation events that occurred during late-night (Fig. 6a) and latemorning (Fig. 6c) time have *single origin* of either land-based or oceanic convection"
(I placed asterisks for emphasis on what I think is dubious). Perhaps the authors would
agree, that evidence of propagation does not necessarily mean that the offshore and
inland rainfall peaks would not exist without this propagation. Perhaps it is equally
plausible that some mechanism(s) favor rainfall in both regions, independent of the
propagation (especially when these peaks are quite separate, as in Fig. 6a)?.

Response #2: We thank the Reviewer for the thoughtful comments. For the first part,
we agree that it is necessary to also point out about the possible effects of gravity
wave to the propagation of rainfall systems. We admit that we missed to mention
that in our previously submitted manuscript. Accordingly, we will add the suggested
references with brief discussions about the possible role of gravity waves in our revised
manuscript.

Secondly, we agree with the Reviewer that *single origin* is not a good wording for
the features that we are trying to emphasize from Figure 6 (a-c). The Reviewer's note
about possible simultaneous occurrence of precipitation events (over land and ocean)
that do not involve propagation is correct and we did not mean to contradict that in our
previous statements. We are trying to revise our manuscript with better wordings.

Comment #3: P. 6 L28: "land-ward shifting of precipitation" Again, Figs. 5 and 7a seem
to suggest that the SCS-CT favors *offshore* rainfall. I do not understand why the
emphasis is placed specifically on land-ward propagation, based on these figures
alone. Colors in Figs. 5 and 6 are saturated, making it difficult to interpret relative
rainfall magnitudes.

Response #3: We thank the Reviewer for the comments. We agree that it is not suffi-
cient to interpret "land-ward shifting of precipitation" only from Figure 5. However, we
should clarify that what we actually wanted to mention is that Fig.5c may confirm that
"land-ward shifting of (oceanic) precipitation" as suggested by Koseki et al. (2012) is
one possible mechanism for the occurrence of morning precipitation over the coastal
region of the West Java.

As to color saturation in Figs.5 and 6, we have tried to add contour lines and also to change the levels of shading to render clearer images.

Comment #4: P.4 L24: "−4.5 m−2" I think you mean m s-1?

Response #4: We thank the editor for the typo correction in P4, L24. The text should read -4.5 msˆ(-1) in our revised manuscript.

Comment #5: P.5 L12–13: Just to clarify, does Fig. 13a,b show (N-days*17-years) samples, or have they been averaged by month? Please indicate in the text.

Response #5: The author thanks the Editor for the comment. We would like to confirm that the AEP, EMP, and LMP samples were constructed from N-days*17-years. We will indicate that more clearly in our revised manuscript.

Comment #6: P.5 L22–23: Could this be due to sampling? I.e., EMP is greatest since it has been averaged over fewer samples than the other categories?

Response #6: We appreciated the Reviewer for the critical comments. In order to respond to that comments, we have tried to put error bars with the mean amplitude plots in Figure 4, so that the variance of the data can be compared for each bin corresponding to the bar charts. The sampling errors ($\pm\sigma/\sqrt{n}$) are plotted as vertical bars shown in the new Figure 4. While it is true that the number of EMP samples is relatively small, overall spread of the amplitudes are (in our opinion) comparable to other groups of data.

Comment #7: P.6 L36: Should be "In Figures 9b and 9c,..."

Response #7: We thank the Reviewer for the typo correction in P6, L36. We have already corrected the text to "In Figures 9b and 9c,..." in our revised manuscript.

Comment #8: Fig. 1: Consider expanding the domain shown to provide a broader context for those less unfamiliar with this region.

[Figure]

Response #8: The authors would like to thank the Reviewer for the comment. We have tried to modify Figure 1 by plotting a larger base map (which is also used in other figures showing maps of wind and SST fields) and put the original Figure 1 as inset. We hope that this can make it easier for readers to understand the geographical context of the studied area.

Please also note the supplement to this comment:
https://www.ann-geophys-discuss.net/angeo-2019-107/angeo-2019-107-AC4-supplement.pdf

[Figure]

Fig. 1. Figure 1. A map showing the Maritime Continent with insert the main area of interest in this study; with coastlines and topographic contours. Red-line bordered polygon (with land masks over the north

Fig. 2. Figure 4. Diurnal distribution of the mean amplitude of the first harmonic components (solid black line) and frequency of occurrence of the peak precipitation time (coloured bar chart) differentiated

[Figure]

**Fig. 3.** Figure 5. Hovmöller diagrams of composite diurnal precipitation; similar to Figure 2 except data are classified by the phase or the timing of peak precipitation (as in Fig. 3): (a) AEP, (b) EMP, and (

[Figure]

**Fig. 4.** Figure 6. Hovmöller diagrams, same as Fig. 5, except averaged data are the 99th percentile (P99) values of the gridded hourly precipitation.

---

## Author Response (AR2)

General comment from **Anonymous Referee #1** (hereafter **Comment**) **#1**:

> *This manuscript investigated the diurnal cycle of precipitation over the northern coast of West Java with a focus on early morning precipitation and influence from SCS-CT and CENS. Well-chosen classification method has clarified the seasonally changing diurnal cycle pattern. Strong correlation between CENS and extreme EMP is also clarified, however, there is no clear link between SCS-CT and variation of diurnal cycle patterns. Therefore, the reviewer would suggest accepting the manuscript for publication after minor revisions.*

Response by Authors (hereafter **Response**) **#1**:

> The authors greatly appreciated the constructive comments from the Reviewer. With regard to the link between SCS-CT and diurnal cycle patterns over the northern coast of West Java, we have actually pointed out as follows:
> a) For all months from November to March, the diurnal precipation pattern is characterised by a main peak during evening to night time. However, in January and February prominent secondary peaks are also observed in the morning time (Figure 2a).
> b) SCS-CT generally prevails in January and February and morning precipitation events over the northern coast of West Java mainly occurred when there was more enhanced SST cooling in the South China Sea (Abstract: Lines 27-28).
> c) In this study, we further classifed morning precipitatin events into early morning and late morning phase groups. We found that the early morning precipitation events were more strongly correlated with CENS, whereas CENS is positively correlated with SCS-CT.
> Therefore,  we can say that SCS-CT has the role to induce morning precipitation over the coastal region of West Java (in agreement with Koseki et al., 2012). SCS-CT also play a role as background condition for the occurrence of CENS  that in turn induced early morning precipitation events.

**Comment #2**:

> *P1, L20: "characterized by seaward (as well as landward) propagation" As is shown in P6, L10, landward propagating oceanic convection is major determinant of EMP in the northern coastal area, though seaward propagation in the inland area in the nighttime may have some linkage. There is a gap in this description and different from the fact in Figs 10 and 11.*

**Response #2**:

> We thank the Reviewer for the comment regarding a gap between depiction of figures and the figures themselves. However, we would like to clarify that Figs. 10 and 11 mentioned by the Reviewer are actually Figs. 5 and 6. The gap as mentioned by the Reviewer seems to refer to direction of propagation in Figs. 5 and 6, which also applies to Figs. 2b-f. Please note that the 24-hourly precipitations data are sorted from 1300 LT to 1200 LT **in the following day**.
>
> Therefore, in those figures, time increases from top to bottom. In this case, seaward as well as landward propagation of precipitation systems could be misinterpreted if time direction is reversed. To minimize such misinterpretation, we have added annotation for time directions and also white  lines with arrows that can guide readers to follow the direction of propagation (following Mori et al., 2004) in Figure 5 (a) - (b).

[Figure]

Figure 5. Hovmöller diagrams of composite diurnal precipitation; similar to Figure 2 except data are classified by the phase or the timing of peak precipitation (as in Fig. 3): (a) AEP, (b) EMP, and (c) LMP. The black dashed line indicates boundary of coastal area. The white solid arrow represents direction of propagation (following Mori et al., 2004), with seaward propagation is symbolized as (α) and landward propagation is written as (β).

[Figure]

Figure 6. Hovmöller diagrams, same as Fig. 5, except averaged data are the 99th percentile (P99) values of the gridded hourly precipitation.

**Comment #3**:

> P4, L24: -4.5m-2 -> m s-1

**Response #3**:

> We thank the Reviewer for the typo correction in P4, L24. We have already changed the unit of wind from:
>
> Previous unit: $-4.5\ \mathrm{ms}^{-2}$
>
> Current unit: $-4.5\ \mathrm{ms}^{-1}$

**Comment #4**:

> P5, L7: *"Figures 2c-f also show weak signals of land-to-sea propagation of precipitation over coastal region during the night-to-morning transition between 2300LT and 0300LT on the following day."* It is hard to recognize land-to-sea propagation in Figs 2e and 2f.

**Response #4**:

> The author thanks the Reviewer for the thoughtful comment. Regarding Figs. 2 (e) and 2 (f), we argued that there is a weak signal of propagation between land and sea based on visual inspection of precipitation intensity contours larger than 0.2 mm h⁻¹, which appear from late afternoon until late night over land, from midnight to early morning over the coastal region, and continue to

connect with that over the sea on the following day. We have modified Figs. 2(b)-(f) by adding contour lines along with the shading in order to make it clearer about such connection between precipitation systems over land and sea.

[Figure]

Figure 2. (a) Climatology of area-averaged (see Fig. 1) diurnal precipitation over the northern coast of West Java, depicted as line plots of 24-hour composite time series in different colours for the months of November (solid black), December (dashed black), January (solid red), February (solid blue), and March (dashed-dotted black) analysed from TRMM MPA-RT dataset of 2000–2016 period. Other panels show the corresponding time-latitude cross sections (Hovmöller diagrams) of the diurnal composites for the months of: (b) November; (c) December; (d) January; (e) February; and (f) March. Dashed vertical black lines in Fig. 2(b-f) denote the latitudes of northern coastal area of West Java. Averaged precipitation rates in mm h$^{-1}$ are shown as shaded contours.

**Comment #5**:
> P5, L11: (see Fig. 1) -> Fig. 2a

**Response #5**:
> We thank the Reviewer for the typo correction in P5, L11. We have already changed the word
> from:

Previous:        (see Fig. 1)
Current:         (see Fig.2a)

**Comment #6**:
> P6, L36: In Figures 8b and 8c, -> In Figures 9b and 9c

**Response #6**:
> We thank the Reviewer for the typo correction in P6, L36. We have already changed the wording
> from:

Previous:        In Figures 8b and 8c
Current:         In Figures 9b and 9c

**Comment #7**:

*P6, L37: "relatively closer towards the equator that indicates the strengthening of the SCS-CT when associated with morning precipitation over the northern coast of Java." The difference between the three figures (9a-c) is indistinguishable.*

**Response #7**:

The authors would like to thank the Reviewer for the constructive comment. We are aware that it might difficult to see significant difference between SCS-CT changes associated with diurnal rainfall pattern of AEP, EMP, LMP which are exhibited in Figs. 9 (a) - (c). We have made a modification to Fig. 9 (presented below) in terms of isotherm contour intervals and colour gradation to make differences between Figs. 9a-9c more distinguishable.

We have also added a new Fig. 10 from results of further analysis of differences in the spatial patterns of SST corresponding to SCS-CT events without CENS (NCENS) and with CENS in the EMP samples. In this new Fig. 10, the composite SST fields of SCS-CT with CENS clearly show wider region of negative anomalies that extend closer to the equator.

[Figure]

Figure 9. Same as Fig. 8, but for SST (solid line contour) and SST anomaly relative to SCS-CT threshold of 26.4º C (shaded). The dark-blue solid rectangle represents the SCS-CT index area (following Koseki et al., 2012).

[Figure]

Figure 10. Same as Fig. 9, but with smaller domain and for composite of EMP samples a) SCS-CT without CENS, b) SCS-CT with CENS.

**Comment #8**:
*P13, Figure 9: The red solid rectangle -> The dark-blue solid rectangle.*

**Response #8**:
We thank the Reviewer for the correction of wrong word in caption of figure 9 (P13). We have already changed the word from:
Previous:        The red solid rectangle
Current:         The dark-blue solid rectangle

General comment from **Anonymous Referee #2** (hereafter **Comment**) **#1**:

> *The study combines 17 years of TRMM satellite and multi-sensor rainfall data, reanalysis datasets, and SST observations to investigate the climatology of diurnal rainfall patterns in the region of northwestern Java, the largest east–west-oriented island in the western Maritime Continent. While the focus region is highly localized, the Maritime Continent is an exceedingly complex region, thus warranting study of potentially unique dynamics in different sub-regions. Further, it is possible that the results extend to other regions in the Maritime Continent, to some measure, though further work would be necessary to determine this.*

Response by Authors (hereafter **Response**) **#1**:

We greatly thank the Reviewer for giving us the insightful comments. We really appreciated the Reviewer's recogniton of our small contribution and its potential extention to further works that will hopefully lead to better understanding of the building blocks of weather dynamics in the Maritime Continent.

**Comment #2**:

> *Background context and propagation mechanisms: A growing body of  studies argue for the likely importance of gravity waves in governing diurnal offshore rainfall propagation, which often manifests at phase speeds faster than the nocturnal land breeze alone can explain, as first highlighted by Mapes et al. (2003). Two studies that argue for this mechanism in the Maritime Continent are Love et al. (2011) and Ruppert and Zhang (2019). It might be useful to note this mechanism since it is likely relevant to the findings.*
>
> *Love, B.S., A.J. Matthews, and G.M.S.Lister, 2011: The diurnal cycle of precipitation over the Maritime Continent in a high-resolution atmospheric model. Q. J. R. Meteorol. Soc., 137, 934–947, doi:10.1002/qj.809.*
>
> *Mapes, B. E., T. T. Warner, and M. Xu, 2003: Diurnal Patterns of Rainfall in Northwestern South America. Part III: Diurnal Gravity Waves and Nocturnal Convection Offshore. Mon. Weather Rev., 131, 830–844, doi:10.1175/15200493(2003)131<0830:DPORIN>2.0.CO;2.*
>
> *Ruppert, J. H., and F. Zhang, 2019: Diurnal Forcing and Phase Locking of Gravity Waves in the Maritime Continent. J. Atmos. Sci., 76, 2815–2835, doi:10.1175/JAS-D19-0061.1.*
>
> *While propagation is clearly evident in some of the panels, text like the following may not be fully justified by the figures and results (P.6 L11–12): "It should also be clear that extreme precipitation events that occurred during late-night (Fig. 6a) and late-morning (Fig. 6c) time have \*single origin\* of either land-based or oceanic convection" (I placed asterisks for emphasis on what I think is dubious). Perhaps the authors would agree, that evidence of propagation does not necessarily mean that the offshore and inland rainfall peaks would not exist without this propagation. Perhaps it is equally plausible that some mechanism(s) favor rainfall in both regions, independent of the propagation (especially when these peaks are quite separate, as in Fig. 6a)?.*

**Response #2**:

    We thank the Reviewer for the thoughtful comments. For the first part, we agree that it is necessary to also point out about the possible effects of gravity wave to the propagation of rainfall systems. We admit that we missed to mention that in our previously submitted manuscript. Accordingly, we have added the suggested references and brief discussions about the possible role of gravity waves in the revised manuscript with the following details:

    P.2 L12-15: *In addition to the sea-land breezes, mechanisms involving gravity waves have also been proposed to explain diurnal convections over land and adjacent seas outside and in the MC (Mapes et al., 2003; Love et al., 2011). Interestingly, phase-locking characteristics between zonally propagating gravity waves and mesoscale convective systems in the MC have recently been confirmed by Ruppert and Zhang (2019).*

    P.8 L18-24: *We have not yet investigated the mechanisms responsible for the propagation of the coastal precipitation, but direct sea/land-breeze effects do not seem to give satisfactory explanation. Moreover, it can be roughly estimated from Figs. 5a and 6a that the speed of the offshore-propagating precipitation systems is around 7 m s-1. Ruppert and Zhang (2019) pointed out that coupling between slow gravity wave mode, with a phase speed of about 7 m s-1, and offshore-propagating convections in the MC has not been well understood. Alternatively, the role of cold pool (e.g., Teo et al., 2011; Trismidianto et al., 2016) may serve as a concrete and testable hypothesis to be the manifestation of the so-called self-replicating mechanism proposed by Mori et al. (2004).*

    Secondly, we agree with the Reviewer that *single origin* is not a good wording for the features that we are trying to emphasize from Figure 6 (a-c). The Reviewer's note about possible simultaneous occurrence of precipitation events (over land and ocean) that do not involve propagation is correct and we did not mean to contradict that in our previous statements. We have rephrased the statements with:

    P.6 L14-19:*The results, which are depicted in Fig. 6, strongly suggest that extreme rainfall events over the coastal region mainly occur between 0100 and 0400 LT and characterised by the existence of propagating systems from both land and ocean (Fig. 6b).  On the other hand, extreme coastal precipitation events that occurred during late-night (Fig. 6a) and late-morning (Fig. 6c)  seem to be originated only from either land-based or oceanic convection. Hence, there are distinct characteristics of propagating systems associated with extreme AEP, EMP, and LMP events over the coastal region. Furthermore, while oceanic convection seems to be major determinant for the occurrences of morning costal precipitation, linkage to land-based precipitation may also be important for the case of EMP.*

**Comment #3**:

    P. 6 L28: "land-ward shifting of precipitation" Again, Figs. 5 and 7a seem to suggest that the SCS-CT favors *offshore* rainfall. I do not understand why the emphasis is placed specifically on land-ward propagation, based on these figures alone. Colors in Figs. 5 and 6 are saturated, making it difficult to interpret relative rainfall magnitudes.

**Response #3:**

We thank the Reviewer for the comments. We agree that it is not sufficient to interpret "land-ward shifting of precipitation" only from Figure 5. However, we should clarify that what we actually wanted to mention is that Fig. 5c may confirm that "land-ward shifting of (oceanic) precipitation" **as suggested** by Koseki et al. (2012) is one possible mechanism for the occurrence of morning precipitation over the coastal region of the West Java.

As to color saturation in Figs. 5 and 6 (presented below), we have tried to add contour lines and also to change the levels of shading to render clearer images. We have also added arrows to more clearly show the (interpreted) direction of propagation of precipitating systems.

[Figure]

Figure 5. Hovmöller diagrams of composite diurnal precipitation; similar to Fig. 2 except data are classified by the phases or the timing of peak precipitation (as in Fig. 3) (a) AEP, (b) EMP, and (c) LMP. The black dashed line indicates boundary of coastal area. The black solid arrows represent direction of propagation (following Mori et al., 2004), with seaward propagation is symbolized by ($\alpha$) and landward propagation is indicated by ($\beta$).

[Figure]

Figure 6. Hovmöller diagrams, same as Fig. 5, except for the 99[th] percentile (P99) values of the gridded hourly precipitation.

**Comment #4:**

*P.4 L24: "−4.5 m−2" I think you mean m s-1?*

**Response #4**:

We thank the Reviewer for the typo correction in P4, L24. The text should read $-4.5 \text{ ms}^{-1}$ in our revised manuscript.

**Comment #5**:

> *P.5 L12–13: Just to clarify, does Fig. 13a,b show (N-days*17-years) samples, or have they been averaged by month? Please indicate in the text.*

**Response #5**:

The author thanks the Reviewer for the comment. We have added more sentences in the revised manuscript with the following details (please note that we also add one reference, El Kenawi et al., 2015):

P.5 L12-16: *The left panels of Fig. 3 show the first harmonic (diurnal) constituent of each sample of 24-hour rainfall time series, which were constructed from hourly data averaged over the coastal region defined in Fig. 1. In this case, the total number of samples that have been analysed is 373, which is not equivalent to the total number of days in all Januari-February months during the 16-year (2001 to 2016) period because time series with diurnal cycle amplitude less than 0.2 mm h$^{-1}$ are excluded to ensure selection of significant rainfall events (see e.g., El Kenawy et al., 2015, for thresholds of moderate rainfall in TMPA data).*

**Comment #6**:

> *P.5 L22–23: Could this be due to sampling? I.e., EMP is greatest since it has been averaged over fewer samples than the other categories?*

**Response #6**:

We appreciated the Reviewer for the critical comments. In order to respond to that comments, we have put error bars with the mean amplitude plots in Fig. 4, so that the variance of the data can be compared for each bin corresponding to the bar charts. The sampling errors ($\pm \sigma/\sqrt{n}$) are plotted as vertical bars shown in the new Figure 4 (below; text added to revised manuscript P.5 L38-39). While it is true that the number of EMP samples is relatively small, overall spread of the amplitudes are (in our opinion) comparable to other groups of data.

[Figure]

Figure 4. Diurnal distribution of the mean amplitude of the first harmonic components (solid black line) and frequency of occurrence of the peak precipitation time (coloured bar chart) differentiated into three groups (same as in Fig. 3): AEP (red), EMP (green), and LMP (blue). The sampling errors ($\pm\sigma/\sqrt{n}$) are plotted as vertical bars.

**Comment #7**:
> P.6 L36: Should be "In Figures 9b and 9c,…"

**Response #7**:
> We thank the Reviewer for the typo correction in P6, L36. We have already corrected the text to "In Figures 9b and 9c,…" in our revised manuscript.

**Comment #8**:
> Fig. 1: Consider expanding the domain shown to provide a broader context for those less unfamiliar with this region.

**Response #8**:
> The authors would like to thank the Reviewer for the comment. We have modified Figure 1 by plotting a larger base map (which is also used in other figures showing maps of wind and SST fields) and put the original Figure 1 as inset. We hope that this can make it easier for readers to understand the geographical context of the studied area.

[Figure]

Figure 1. A map showing the Maritime Continent and its surrounding area. The main area of interest in this study is depicted as inset; with coastlines and topographic contours. Red-line bordered polygon (with land masks over the northern coastal plain of West Java) defines the area for spatial averaging of gridded precipitation data. Blue-dashed rectangle defines the area for longitudinally (105.5– 108.5º E)  averaged precipitation in Hovmöller analysis of Fig. 2.

**Comment by Topical Editor**:

*Both reviewers suggest minor revisions, and I would very much appreciate to receive a revised manuscript taking into account the reviewers' suggestions. In case the authors can easily obtain temporally resolved observations of near-surface air temperature at a nearby "inland" station, I would be interested to see how symmetric the average daily cycle of near-surface air temperature is in this region. I wonder whether a higher warming rate during daytime may contribute to the sharper precipitation maximum during daytime. Perhaps this could be a topic for an idealized model sensitivity study in the future.*

**Response by Authors:**

We are thankful to the Editor for thoughtfully giving us the suggestions. We have seriously considered all reviewer's comments to improve our manuscript. Herewith, we would also like to respond to additional comments from the Editor:

a) We have already obtained the hourly surface temperature data from Indonesian Meteorological, Climatological, and Geophysical Angency (BMKG) station at Soekarno-Hatta International Airport for the months of NDJFM during 2013 to 2016.

b) We have anlysed the composite diurnal cycle of the surface temperature for each month as shown in the Figure below.

c) In general, average surface temperatures in January and February are lower than those in November, December, and March. While average temperature of March is lower than those of November and December, coastal rainfall has conversely the sharpest diurnal peak during March (see Fig. 1 in our manuscript).

d) Diurnal variations of temperature and its correlation with sea/land-breeze, and also convections, over Jakarta area have been documented by previous works of, among others, Hadi et al. (2002) and Araki et al. (2006).  It might be important to note that, as pointed in these previous works, "sea-breeze" during rainy season has very different timing (inland penetration occurs much earlier) with no return flow in the boundary layer (lower troposphere).  On the contrary,  well defined sea-breeze circulations associated with differential heating between land and adjacent ocean have been clearly observed.

e) We have decided not to include this discussions in our present work because we think it will be a subject of different research.

[Figure]

Figure Caption: Diurnal cycle of surface temperature average in Soekarno-Hatta station (Jakarta area) based on hourly data of BMKG during NDJFM of 2013-2016.

**References:**

Araki, R., Yamanaka, M. D., Murata, F., Hashiguchi, H., Oku, Y., Sribimawati, T., Kudsy, M., and Renggono, F.: Seasonal and interannual variations of diurnal cycles of wind and cloud activity observed at Serpong, West Java, Indonesia, J. Meteor. Soc. Japan, 84A, 171–194, 2006.

Hadi, T. W., Horinouchi, T., Tsuda, T., Hashiguchi, H., and Fukao, S.: Sea-breeze circulation over Jakarta, Indonesia: A climatology based on boundary layer radar observation, Mon. Wea. Rev., 130, 2153–2166, 2002.